# Explaining Machine Learning Models Based on Conditional Expected Prediction

## Abstract

Complex machine learning models are increasingly used across various fields, but gaining insight into their decision-making processes remains a challenge. Numerous explanation methods have been developed in recent years, aiming to clarify how these models work from different perspectives. Recent studies have shown that some of these explanation methods may produce misleading results, particularly when features are correlated, such as with background noise or correlated features lacking relevant information related to the target. Among different methods, those based on conditional expected prediction have demonstrated greater robustness to such features. However, applying these methods requires knowledge of the conditional distribution, i.e., the distribution conditioned on a specific feature, which is challenging to estimate. Current approximation methods require additional assumptions about the data and models. We propose a global model-agnostic explanation method based on conditional expected prediction. Our method approximates conditional expected predictions through data partitioning and kernel-based methods, eliminating the need for additional assumptions. We validate our method using synthetic data and open-source EEG data, and the results demonstrate that it is significantly less affected by correlated features.

## 1 Introduction

Machine learning has demonstrated remarkable power to solve complex problems in several domains by analysing large datasets and uncovering hidden information. Although these complex models are powerful tools, their decision-making processes, such as selecting which features to use, often remain a "black box" for humans. This lack of transparency challenges the trust of these models in practical application. In response to this issue, research on interpretability and explainability has gained significant attention.

Various methods have been proposed to explain a model from different perspectives. Explanations can be made through visualisation, which displays the crucial areas in an image (Shrikumar et al., 2017; Kindermans et al., 2017), or shows how alterations in feature values affect the outcomes (Friedman, 2001; Apley & Zhu, 2020). The counterfactual method (Mothilal et al., 2020; Stepin et al., 2021) aims to determine how features change when the model alters its prediction for a specific instance. A critical aspect of model explanation is measuring feature contributions. Various efforts have been made to address this issue, like permutation feature importance (Fisher et al., 2019), Shapley-based methods (Lundberg & Lee, 2017; Covert et al., 2020).

However, the model explanation methods may be misleading when features are correlated. Various model explanation methods rely on the feature independence assumption, which may not hold in real applications (Zhao et al., 2024; Herbinger et al., 2023), which may potentially introduce bias. In Budding et al. (2021), researchers intentionally added classification-irrelevant artefacts to the MRI images. Models are trained using the altered datasets, along with the implementation of various explanation methods. The results indicate that these methods are influenced by the MRI images themselves, highlighting pixels associated with the artefacts.

In Wilming et al. (2022), synthetic datasets with linear relationships are built to test the model explanation methods. The informative features are manually adjusted to ensure that they correlate with another feature, independent of the target variable. The results indicate that most explanation methods can produce misleading outcomes, whether at the local or global level. A similar study

was conducted (Wilming et al., 2023) in which this case of feature correlation is considered as a suppressor variable, and a theoretical discussion is presented under the assumption of linear separability and the use of linear classifiers. Methods based on conditional expectations, like the Feature Importance Ranking Measure (FIRM) (Zien et al., 2009), can be less affected by correlated features, like suppressor variables, which are correlated with other informative features but provide less or no information related to the target.

FIRM provides a global-level feature contribution based on the variation of the conditional expected prediction, which is the expected prediction conditioned on the feature under explanation. One challenge with this method is that it requires access to this conditional distribution to calculate the results, which is usually not available, particularly for complex datasets. In Zien et al. (2009), the authors proposed estimation methods for linear cases by assuming that the data are in Gaussian distribution. In Haufe et al. (2014), researchers proposed a method that can be seen as a special case used for linear parametric models, which is not applicable to nonparametric models.

This paper makes three main contributions.

1. Proposes model-agnostic methods for measuring global feature contributions.

2. Introduces two efficient approaches for estimating global feature contributions: ApprFIRM-quantile, which uses quantile partitions, and ApprFIRM-kernel, which adopts kernel estimation.

3. Conducts extensive experiments on synthetic and real EEG data, demonstrating that the proposed methods are more robust than existing approaches when handling data with correlated features, such as suppressor variables.

## 2 RELATED WORK

Multiple studies indicate that correlated features can lead to misleading results from explanation methods(Wilming et al., 2022; 2023; Apley & Zhu, 2020; Strobl et al., 2008; Molnar et al., 2024). One form of misleading is caused by suppressor variables. This kind of variable can improve the predictive power of other variables while showing little connection or no direct contribution to the target. Suppressor variables, initially studied in regression analysis (Conger, 1974; Friedman & Wall, 2005), in which these variables present no correlation with the target but indeed enhance model performance. Recent studies have also explored suppressor variables in contexts beyond regression (Pandey & Elliott, 2010; Lynn, 2003). In Kim (2019), researchers explore the concept of suppressor variables from a causality perspective, offering a thorough analysis of the suppression effect across various causal structures. They indicate that a suppressor variable is similar to the instrumental variable in the context of causal inference. In causal inference, instrumental variables are essential for estimating causal effects in the presence of unobserved confounding bias (Wu et al., 2022). These unobserved confounding biases are caused by the unobserved confounder variable, which influences both the feature being analysed and the target variable. This can lead to a misleading effect between the feature being analysed and the target. As noted in some studies (Wooldridge, 2016; Steiner & Kim, 2016), these variables could be harmful when included in the analysis because the hidden bias can be amplified. While identifying these suppressor variables may not be essential from a model performance perspective, it is important in the context of model explanation to understand whether and how much this instrumental variable relates to the target variable.

In Wilming et al. (2023; 2022), researchers conduct theoretical and experimental analysis of the explanation methods in linear classification tasks involving suppressor variables. Among the state-of-the-art explanation methods being tested, most are significantly influenced by the suppressor variable. However, the FIRM method demonstrates a reduced sensitivity to these influences and produces results that are less affected. The FIRM (Zien et al., 2009) is a model-agnostic approach designed for generating global-level feature contribution explanations. This method analyses changes in the model's conditional expectations, which can reveal how predictions shift in response to specific features and better handle feature-correlated cases. However, estimating conditional expectations requires accessing the conditional distribution, which is challenging, especially when facing high-dimensional data. The research presents several approximation methods based on the assumption of Gaussian distributed data or linear models. In Haufe et al. (2014), a comparable methodology is introduced from the data generation perspective. Nevertheless, a notable limitation of both meth-

ods is their applicability solely to linear models. Furthermore, these approaches necessitate model parameters that are applicable only to specific parameter models. In Zhang et al. (2024), researchers take a further step by extending this method into kernel space, implementing it to explain kernel-based SVM models.

Estimating conditional expectation presents significant challenges not only within the framework of the FIRM method but also extends to various other methods. A common approach is to assume features are independent, making it easier to implement Monte Carlo estimation. However, this assumption can introduce biases, such as the extrapolation problem (Molnar et al., 2020), where the sampling range exceeds the actual data distribution. In Apley & Zhu (2020), a visual approach for explaining models is presented, which also requires estimating local conditional predictions. To tackle this issue, they first partition data into small subgroups and assumed that samples in these subgroups fulfil the associated conditional distribution. This method is straightforward to implement with low computation cost. However, these approaches require predefined partitions or sample neighbours, which can affect the efficiency of the explanation results. A method that can automatically partition the data through tree models is proposed in Molnar et al. (2024). However, the performance of this method declines when applied to continuous features, such as those that are linearly correlated. Several approaches have been proposed for approximating conditional samples using alternative models, such as variational autoencoders (Olsen et al., 2022; 2023) and deep learning models (Chamma et al., 2024). However, one issue with these methods is that the performance of these alternative models can significantly impact the results of the explanations. Additionally, training these models typically requires large datasets.

## 3 METHOD

Various explanation methods utilize marginal distribution to estimate the feature importance, which may lead to extrapolation problems (Molnar et al., 2024). This issue occurs when using samples derived from the marginal distribution that does not accurately reflect the actual data distribution. Those samples may be unrealistic in terms of the actual data distribution. This problem can be mitigated by employing the conditional distribution instead of the marginal distribution. However, directly calculating conditional expected scores can be challenging, as obtaining conditional distributions is often infeasible due to the curse of dimensionality and the limited amount of data. Due to this challenge, we introduce two approximation methods to approximate the conditional expected prediction; one is based on quantile partitions, and another is based on a kernel estimator. The feature importance score is obtained based on these approximated results.

**Notation** Consider a dataset $(\mathbf{X}, \mathbf{Y})$, where $\mathbf{X} \in \mathbb{R}^{n \times d}$ is a matrix of $n$ samples and $d$ features, and $\mathbf{Y} \in \mathbb{R}^n$ is the corresponding target vector. $\mathbf{x}^i \in \mathbb{R}^d$ represents the $i$-th sample. $f(\mathbf{x}) : \mathbb{R}^d \to \mathbb{R}$ represents the model. The $s$-th feature is represented as $\mathbf{X}_s$, and the value of the $s$-th feature for the $i$-th sample is represented as $x_s^i$, while $\mathbf{x}_{-s}^i$ represents the feature values of the $i$-th sample excluding the $s$-th feature.

---

**Algorithm 1:** Algorithm for partition-based approximation of feature importance score

**Input:** data matrix $\mathbf{X}$, the number of intervals: $K$, the $s$-th feature: $\mathbf{X_s}$, model: $f(\mathbf{x})$

Divide data $\mathbf{X}$ into $K$ intervals based on quantiles of feature $\mathbf{X}_s$. Let $\{\mathbf{x}^j\}^k$ be the $j$-th sample in the $k$-th quantile of feature $\mathbf{X}_s$;

**for** $k = 1$ **to** $K$ **do**

$\quad \mathbf{CScore}_s^k = \frac{\sum_{\mathbf{x}^j \in \{\mathbf{x}^j\}^k} f(\mathbf{x}^j)}{\text{The number of samples in } k\text{-th partition}}$;

**end**

$\mathbf{ApprFIRM}(\mathbf{X_s}) = std(\{\mathbf{CScore}_s^1, ..., \mathbf{CScore}_s^k\})$;

---

### 3.1 APPROXIMATION BASED ON QUANTILE PARTITION

We propose a method to approximate the conditional expected prediction through data partitioning, which is inspired by (Apley & Zhu, 2020). In this method, the conditional expected predictions of

---

**Algorithm 2:** Algorithm for kernel-based approximation of feature importance score

---

**Input:** data matrix $\mathbf{X}$, number of samples $n$, the $s$-th feature: $\mathbf{X}_s$, bandwidth: $\sigma$, model: $f(\mathbf{x})$

Calculate the variance of feature $\mathbf{X}_s$ as $var(\mathbf{X}_s)$;

**for** $i = 1$ **to** $n$ **do**

    **for** $j = 1$ **to** $n$ **do**

        Calculate the normalized distance: $\mathbf{d}(x_s^i, x_s^j) = \sqrt{\frac{(x_s^i - x_s^j)^2}{var(\mathbf{X}_s)}}$;

        Calculate the weight: $w_{ij} = \mathbf{exp}\left(-\frac{\mathbf{d}(x_s^i, x_s^j)^2}{\sigma}\right)$;

        Replace the $s$-th feature value of $j$-th sample $x_s^j$, with the feature value of $i$-th sample $x_s^i$;

        Calculate the prediction using the replaced sample: $f(\mathbf{x}_{-s}^j, x_s^i)$

    **end**

    Calculate the conditional score for $i$-th sample: $\mathbf{CScore}_s^i = \frac{\sum_{j=1}^n w_{ij} f(\mathbf{x}_{-s}^j, x_s^i)}{\sum_{j=1}^n w_{ij}}$;

**end**

$\mathbf{ApprFIRM}(\mathbf{x_s}) = std(\{\mathbf{CScore}_s^1, ..., \mathbf{CScore}_s^n\})$;

---

feature $\mathbf{X}_s$, represented as $\mathbf{CScore}_s^k$, are measured at the $k$-th quantile partition $\{\mathbf{x}^j\}^k$. The process begins by partitioning the data based on the values of $\mathbf{X}_s$, which involves determining $K$ quantiles of this feature. The samples are then divided into partitions according to these quantiles, with the subset of samples belonging to the $k$-th quantile partition denoted as $\{\mathbf{x}^j\}^k$. Predictions are then made for the samples in the $k$-th partition using the model $f(\mathbf{x})$. The conditional expected prediction for the $k$-th partition of feature $\mathbf{X}_s$, denote as $\mathbf{CScore}_s^k \in \mathbb{R}$, is approximated by averaging the predictions within this partition. It should be noted that the data samples within the same partitions are assumed to have the same conditional distributions. Our algorithm, based on data partitioning, is summarised in Algorithm 1.

### 3.2 APPROXIMATION BASED ON KERNEL ESTIMATOR

The quantile partition based method is intuitive and computationally efficient. However, when the sample size is small, the resolution of results may be compromised due to the need for multiple partitions. To address this issue, we propose an alternative approach based on kernel estimators inspired by (Aas et al., 2021), which are less affected by the sample size but require higher computational costs compared to the quantile partition-based method.

Instead of measuring the conditional expected score at each partition, the kernel-based method measures the conditional expected score, $\mathbf{CScore}_s^i \in \mathbb{R}$, at each instance $\mathbf{x}^i$ for feature $X_s$.

Firstly, the distance between the feature value $x_s$ of the current sample $x_s^i$ and each of the other samples $x_s^j$ is measured as $\mathbf{d}(x_s^i, x_s^j) = \sqrt{\frac{(x_s^i - x_s^j)^2}{Var(X_s)}}$, where $j = 1, 2, ..., n$ and $j \neq i$.

The distance is normalised by the variance of the feature $\mathbf{X}_s$, which makes the distance less affected by different feature ranges. Then, the sample weights based on the above distance measures are generated through the Radial Basis Function (RBF) kernel as $w_{ij} = \mathbf{exp}(-\frac{\mathbf{d}(x_s^i, x_s^j)^2}{\sigma})$. These weights measure the similarity between the current sample and other samples from the perspective of feature $\mathbf{X}_s$. The coefficient of $\sigma$ determines how sensitive the weights are to nearby samples.

The next step involves calculating predictions for the modified data samples. The data sample $x^j$ is modified by replacing the feature value $x_s^j$ with the feature value from the current sample $\mathbf{x}^i$, which is represented as $x_s^i$. The modified sample is denoted as $(\mathbf{x}_{-s}^j, x_s^i)$ and its prediction is $f(\mathbf{x}_{-s}^j, x_s^i)$. The conditional expected prediction for sample $\mathbf{x}^i$ is the weighted sum of these predictions, with the associated weights determined in the previous steps. The equation is $\mathbf{CScore}_s^i = \frac{\sum_{j=1}^n w_{ij} f(\mathbf{x}_{-s}^j, x_s^i)}{\sum_{j=1}^n w_{ij}}$

The kernel-based algorithm is shown in Algorithm 2.

### 3.3 FEATURE IMPORTANCE

After obtaining the conditional expectation score, the approximated global feature importance score for feature $\mathbf{X}_s$ is obtained by quantifying the variation of the scores. Standard deviation (std) is introduced to measure the variation. A higher feature importance score indicates a significant change in the expected prediction as the feature value varies. This suggests that the feature holds relevant information. In contrast, a lower feature importance score means that changes in the feature lead to minimal changes in the expected predictions, indicating that the feature contains little relevant information. In summary, the final result for feature $\mathbf{X}_s$, the $\mathbf{ApprFIRM(X_s)}$ can be described as:

$$\mathbf{ApprFIRM(X_s)} = \text{Std}(\mathbf{CScore}_s), \text{ where } \mathbf{CScore}_s = \mathbb{E}[f(\mathbf{X})|\mathbf{X}_s = x_s] \tag{1}$$

## 4 EXPERIMENT

A major challenge in verifying model explanations is the absence of known ground truth in most real-world datasets. To address this, synthetic datasets are created to simulate conditions with pre-defined ground truths. Our method is first tested on these synthetic datasets—covering both linear and nonlinear cases—then tested on an open-source real EEG dataset. Three machine learning methods are selected to represent a diverse range of models: Support Vector Machine (SVM), Random Forest (RF), and a 3-layer Neural Network (NN). State-of-the-art feature importance methods are used for comparison, including Local Interpretable Model-agnostic Explanation (LIME) (Ribeiro et al., 2016), Shapley Additive Explanations (SHAP) (Lundberg & Lee, 2017), and Permutation Feature Importance (PFI) (Fisher et al., 2019). LIME scores are calculated for every feature, while default settings are used for PFI and SHAP. All experiments are conducted on a desktop with an Intel i7 9700K CPU and 32 GB RAM. The example code can be found at https://github.com/Zhmq117/ApprFIRM/.

### 4.1 SYNTHETIC DATA

The simulation task is in a binary classification setting. In each scenario, 100 synthetic datasets, each with 2000 samples and 5 features, are used. The classification information is provided by features $\mathbf{x}_1$ and $\mathbf{x}_4$, while feature $\mathbf{x}_5$ is non-informative and independent in both linear and nonlinear scenarios. In the linear scenario, features $\mathbf{x}_2$ and $\mathbf{x}_3$ are correlated with $\mathbf{x}_1$ from opposite directions. In nonlinear case, $\mathbf{x}_2$ is correlated with $\mathbf{x}_1$, whereas $\mathbf{x}_3$ is an independent feature. In a linear scenario, the SVM model uses a linear kernel, while in a nonlinear scenario, it uses an RBF kernel. For the neural network models, each hidden layer consists of 10 neurons using the ReLU activation function. A softmax layer is used in the final layer. The explanation results are obtained from a separate test set.

Datasets are generated to simulate suppression cases in classification settings, in a similar way as in the works of Wilming et al. (2022; 2023). The features exhibit correlation with other non-informative features, which arises from either direct correlation or overlapping signals.

#### 4.1.1 LINEAR

The dataset is generated through the multivariate Gaussian distribution framework. It includes five features, of which features $\mathbf{x}_1$ and $\mathbf{x}_4$ contain class-related information while the others do not. This differentiation is achieved by utilizing distinct mean vectors for the two classes when generating the data. Specifically, the feature values of $\mathbf{x}_1$ and $\mathbf{x}_4$ are assigned a value of 1 for the positive class and -1 for the negative class, while all other features are initialized to 0. Additionally, feature $\mathbf{x}_1$ exhibits a positive correlation with $\mathbf{x}_2$ and a negative correlation with $\mathbf{x}_3$. Features $\mathbf{x}_4$ and $\mathbf{x}_5$ are independent of feature $\mathbf{x}_1$. These dependencies are established by adjusting the covariance matrix of the Gaussian distribution.

#### 4.1.2 NONLINEAR

The data consists of three parts: signal, overlapped distractor, and random noise. The signal part contains class-related information (features $\mathbf{x}_1$ and $\mathbf{x}_4$), While the distractor part (between features

$x_2$ and $x_1$) represents the overlapped class-irrelevant information. The signal is overlapped at $x_1$, i.e., the sample value of $x_1$ contains both information that comes from the signal part and the over-lapped distractor part, as well as random noise. This setting introduces a correlation between $x_1$ and $x_2$. $x_2$ can be seen as a suppressor variable since it contains no class-related information but can potentially be used for denoising. The classes are defined by setting features $x_1$ and $x_4$ as 1 or -1 for the positive class and 0.25 or -0.25 for negative class, respectively. This setting introduces the nonlinear relationship. The distractor part is a fixed vector multiply $\rho$, sampled from standard normal distribution $N(0, 1)$. Random noise parts are sampled from multivariate Gaussian distribution with zero means $N(0, \Sigma)$. To sample the covariance matrix $\Sigma$, we begin by generating a 5 by 5 matrix representing the covariance matrix of the multivariate Gaussian distribution that is randomly sampled from a standard normal distribution. Subsequently, we compute the dot product of this matrix, which guarantees that the resulting covariance matrix is positive semi-definite. To ensure standardization, the diagonal elements of this matrix are normalized by dividing each element of the covariance matrix by the product of the standard deviations of the corresponding rows and columns. The resulting normalized matrix can thus be interpreted as a correlation matrix.

All signal, distractor, and noise sections will be normalised by their respective Frobenius norms. The proportion of signal, $coef_s$ is set to 0.3, while proportion of distractor and noise $coef_d = coef_n = (1 - coef_s)/2$. The overall data is: $\mathbf{X} = \mathbf{coef_s} * \mathbf{signal} + \mathbf{coef_d} * \rho * \mathbf{distractor} + \mathbf{coef_n} * \mathbf{noise}$

## 4.2 REAL DATA

To evaluate the effectiveness of our method on real data, we validated it using open-source EEG data (Wakeman & Henson, 2015). This dataset is collected during a visual task focused on face perception. During the data collection process, participants were presented with images of famous faces, unfamiliar faces, and scrambled faces, which are organized into three categories. The dataset involves sixteen participants, each contributing approximately 300 samples per class. For our validation, we specifically focused on the famous faces and scrambled face class.

**Preprocessing** The preprocess involves the application of a bandpass filter within the frequency range of 1 Hz to 40 Hz with windowed sinc Finite Impulse Response (FIR) filters. This procedure effectively mitigates noise originating from other activities occurring at other frequencies. Subsequently, the data is re-referenced utilizing the average reference method. Channels that do not directly capture brain signals, such as those for Electrocardiogram (ECG) and Electrooculography (EOG), are excluded from the dataset. To enhance computational efficiency, the signal undergoes downsampling and segmentation in accordance with the event file associated with the dataset. Each segment represents a sample that includes 500 ms before the images are presented and 1000 ms afterwards. Baseline correction is applied during the time window from 500 ms to 0 ms before the images are displayed. This step helps to reduce the effects of temporal drifts. A total of 70 channels, or electrodes, are retained as sensor-level features.

For classification tasks, we selected two time intervals: Interval 1 spans from 80 ms to 120 ms, representing the P100 component (Boutros et al., 1997; Earls et al., 2016); and Interval 2 ranges from 150 ms to 190 ms, which corresponds to the N170 component (Brunet, 2023; Hinojosa et al., 2015). The signal within the selected time interval is averaged as a feature for the models.

After preprocessing, the experiments for synthetic data are conducted separately, using RF, RBF-SVM, and Neural Network models. Unlike synthetic data, the EEG data are normalised using a standard scaler for model training and explanation.

## 5 RESULTS

As for the convenience of comparing different results, all scores are normalised to 0 and 1 using the min-max scaling method.

## 5.1 SYNTHETIC DATA

The results are shown in a box plot to indicate the effectiveness and stability of the results. All results are min-max scaled between 0 and 1 for the convenience of presentation. As the data generation

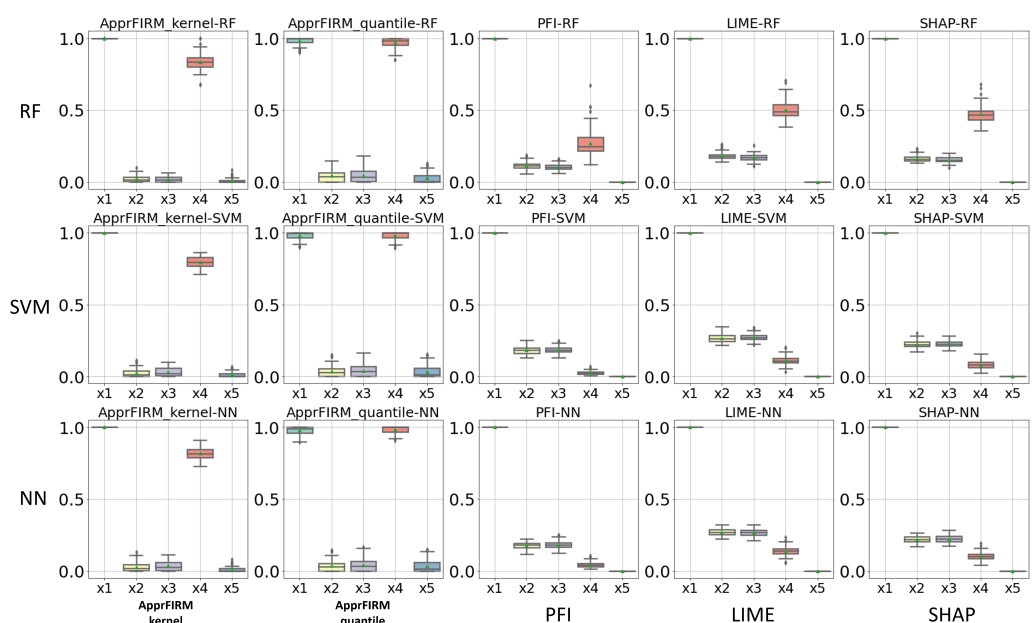

Figure 1: Results of the linear scenario. All results are min-max scaled between 0 and 1 for the convenience of presentation. Feature $x_1$ and $x_4$ contain class-related information and should receive high scores. All methods give $x_1$ highest scores and $x_5$ the lowest. Our approaches, both based on kernel estimator and quantile partitions, successfully assign a high score to feature $x_4$. The other 3 methods assign significant lower scores except in RF models.

procedure shows in the previous section, the features $x_1$ and $x_4$ contain class-related information and should receive high scores, while the other 3 features should receive low scores.

The results for the synthetic data experiments are shown in Figure 1 for the linear scenario. Our approaches, both based on kernel estimator and quantile partitions, successfully assign a high score to both features $x_1$ and $x_4$ and a low score to the other three features. While the kernel-based method demonstrates less stability in scoring informative features, it is more effective at suppressing the scores of class-irrelevant features compared to the quantile partition-based method. The other 3 methods also give higher scores to $x_1$ but struggle to identify feature $x_4$, with the exception of RF models. Additionally, these methods assign relatively higher scores to $x_2$ and $x_3$ across all experiments with different models, indicating that they are influenced by the class-irrelevant features that are correlated with $x_1$.

Figure 2 illustrates the results for the nonlinear scenario. All methods exhibit less stability compared to those in the linear scenario. Our methods successfully identify the informative features $x_1$ and $x_4$, while assigning relatively low scores to the other features. The kernel-based methods exhibit instability in their scores for $x_1$ compared to the quantile partition-based method. However, the score for $x_1$ remains significantly higher than those of other class-related features. The other three methods can roughly identify the informative features, but the results are notably unstable in experiments involving SVM and NN models, particularly regarding the informative features $x_1$ and $x_4$. Additionally, PFI and SHAP assign relatively high scores not only to $x_2$, which is correlated with $x_1$, but also to the other two independent features. Among all comparable methods, LIME is less affected by the correlated feature; however, its results still display significant instability compared to our methods.

Additional experimental results of different combinations of the correlating coefficient and sample amount are shown in the appendix.

## 5.2 EEG DATA

The results are shown in Figure 3 in topography format. Topography is a visualization tool commonly used to present brain electrical activity on the scalp. The highlighted areas indicate the

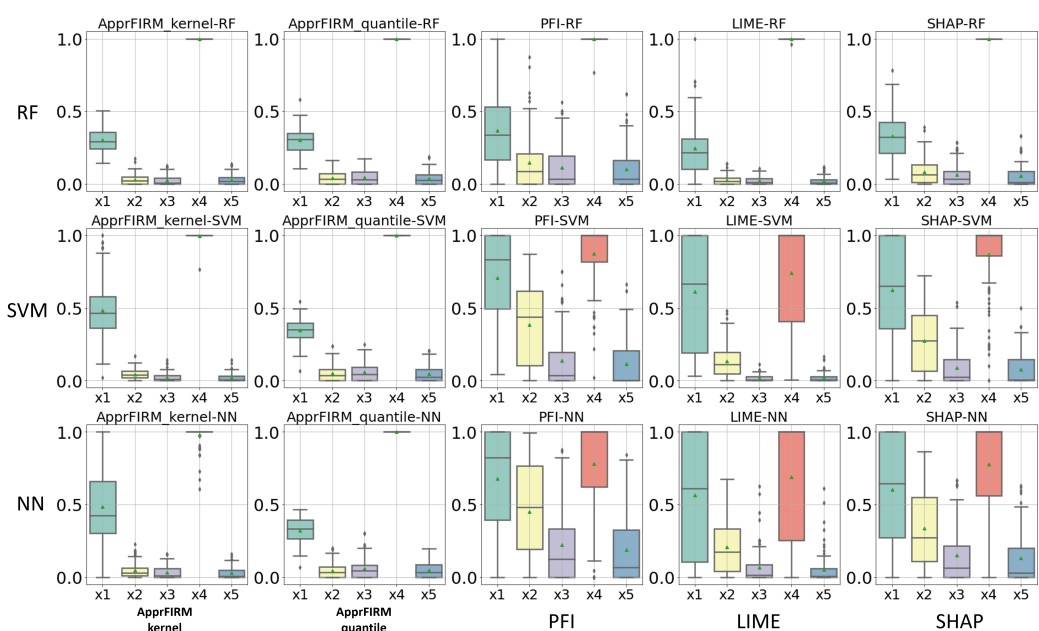

Figure 2: Results of the nonlinear scenario. All results are min-max scaled between 0 and 1 for the convenience of presentation. Feature $x_1$ and $x_4$ contain class-related information and should receive high scores. All method shows less stable results than in linear case. Our method correctly assign high scores to both $x_1$ and $x_4$ and the variance of the kernel-based method is larger than the quantile-based method. However, the variance of our methods is smaller than the other 3 methods. The other 3 methods are influenced by other features and failed to assign a low score to $x_2$, except in the RF model.

regions that are active during the studied event. For comparison purposes, the results are first taken in absolute value and averaged among 16 participants, and then rescaled to a range between 0 and 1 using a min-max scaler.

As mentioned in the previous section, two time intervals have been selected. These intervals correspond to two signal components, the P100 and N170, related to the visual face recognition task, which have been reported in many previous studies (Brunet, 2023; Boutros et al., 1997; Earls et al., 2016; Hinojosa et al., 2015; Maurer et al., 2008).

The results for the first interval (the P100 component) are presented in Figure 3-A. The P100 component is typically detected around 100 ms after the stimulus, indicating the early processing of visual stimuli. It is sensitive to various low-level properties of visual inputs (Negrini et al., 2017; Rossion & Jacques, 2008). Channels located at the back of the head primarily measure signals from the occipital cortex, typically in both the left and right hemispheres. However, as findings in the previous physiological study (Negrini et al., 2017), the observed signal differences may be asymmetric, with the right hemisphere often recording larger signal differences. As demonstrated in the results presented in Figure 3-A, our methods, both kernel-based and quantile partition-based method, identify active areas that are better consistent with findings in previous studies compared with other methods. However, the explanation results of our explanation methods in experiments with NN models involve more area, especially for the quantile partition based method. One potential reason is the limited samples. Each test set used to calculate the explanation results contains approximately 150 samples, which may result in a decrease in accuracy as model complexity increases. LIME and SHAP identify a similar active areas for the Random Forest (RF) and Neural Network (NN) models but highlight different channels when applied to RBF-SVM models. In contrast, Permutation Feature Importance (PFI) did not identify any meaningful areas when applied to the RF models.

The results for the second interval, which corresponds to the N170 component, are presented in Figure 3-B. The N170 component is a signal difference occurring approximately 170 ms after the stimulus in the face recognition study, linked to high-level cognitive processes such as face detection

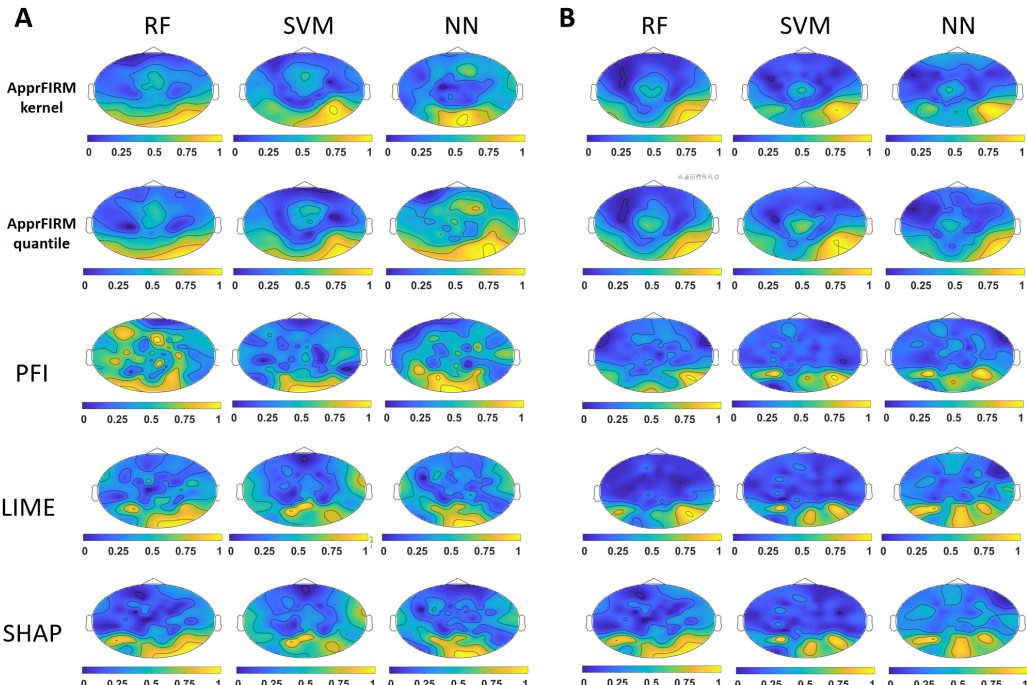

Figure 3: The figure shows the results of EEG data. All results are averaged and min-max scaled between 0 to 1 for the convenience of display. Figure A shows the results of interval 1, which contains signals between 80 - 120 ms. Figure B shows the results of interval 2, which contains signals between 150 - 190 ms.

(Rossion & Caharel, 2011). Typically, this component is recorded from channels located over the posterior temporal and occipitotemporal regions on the lower part of the back head in both hemispheres except the mid-back head area (Caharel & Rossion, 2021). Although the component can be detected in both hemispheres, the signal may be asymmetric, with a more significant reaction often observed in the right hemisphere's occipitotemporal region Rossion & Caharel (2011). The results in the figure demonstrate that our method emphasizes both hemispheres of the occipital-temporal region, focusing on the right hemisphere, but the left hemisphere receives comparatively less emphasis. In contrast, areas highlighted by the other three methods not only include the occipital-temporal region but also, to different extents, including the mid-back area, which is the occipital region. Our methods are more in line with previous physiological studies, suggesting greater consistency and reliability.

## 6 CONCLUSION AND LIMITATION

This paper introduces model-agnostic explanation methods that leverage the demonstrated strengths of FIRM to provide more accurate explanations for correlated features than existing methods. We present two distinct approximation approaches to address the challenges of estimating conditional expected predictions. These methods evaluate how conditionally expected predictions of the model vary as individual features change. Since the scores are approximated under a conditional distribution, the extrapolation is avoided. Moreover, our methods are less affected by suppressor variables. Our methods have been validated using both synthetic data and open-source EEG data. A limitation of the proposed methods is that the kernel based method incurs higher computation costs as sample size increases. In contrast, the partition based method is computationally efficient but may be less effective for complex distributions involving discrete features. Future research could explore partitioning strategies that are more effective for discrete data.

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
