# OpenReview forum: "Explaining Machine Learning Models Based on Conditional Expected Prediction"
_ICLR.cc/2026/Conference — ICLR 2026 Conference Withdrawn Submission_

### Official Review · Reviewer_WFRL · 2025-10-18

**Soundness:** 2
**Presentation:** 2
**Contribution:** 1
**Rating:** 2
**Confidence:** 4

**Summary:**

The paper studies feature-importance (FI)-based model explanation methods under correlated features, where suppressor variables can appear important despite having little direct effect on the target because their influence is mediated through other variables. While importance measures based on condition expectation such as FIRM mitigate this issue, the setting is mainly limited to linear/Gaussian models due to the difficulty of estimating conditional expectations in high dimensions. The authors propose model‑agnostic, global FI methods that approximate FIRM for complex, nonparametric models via two estimators: ApprFIRM‑quantile (quantile partitions) and ApprFIRM‑kernel (kernel estimation). They claim these estimators yield more reliable importance under feature correlation—especially with suppressors—than existing approaches.

**Strengths:**

* The proposed method apply post‑hoc to arbitrary predictors.
* The paper is generally clearly written and the estimation ideas are well motivated.
* Two practical approximations (quantile and kernel) offer complementary trade‑offs for conditional expectation estimation.

**Weaknesses:**

The key issue with this paper is the problem setup. While the problem of correlated features is reasonable in the usual context of data analysis, it does not really make sense to me in the context of model explanation. The paper equates a “good explanation” with recovering the true data‑generating features. In model explanation, however, fidelity is typically defined with respect to what the trained model actually uses—which may differ from the ground truth. Experimental results only show the alignment with the true features rather than features that affect model predictions/performance.  It can be that a feature is truly an suppressor but not treated so and wrongfully exploited by the model. A good explanation method in this case should arguably one that can demonstrate such blind spots of black-box models for users to understand how reliable the models are when used in practice.

The importance score is not described in enough detail to compare meaningfully with alternatives (e.g., Shapley/SHAP, conditional SHAP, permutation FI). The paper asserts FIRM is “less sensitive” to correlation, but how the formulation reflects so is under‑explained.

Other important aspects that are also left undiscussed. See the questions below.

**Questions:**

1/ How is ApprFIRM more ideal than existing importance measures like Shapley/SHAP (marginal vs. conditional) or permutation FI? What properties (e.g., symmetry, monotonicity, boundedness) does it preserve or trade off?

2/ FIRM is based on standard deviation. To what extent does the measure truly reflect feature attribution rather than estimation noise?

3/ As far as I understand, Algorithm 1 & 2 only deal with regression settings (real-valued target variable). However, the authors reported experiments on classification tasks (Line 308), but no exact details on how ApprFIRM‑quantile/kernel extend to classification.

4/ The main claim of the proposed method is to deal with high-dimensional data. The only related experiment is the EEG study, but no exact details on how many features are analyzed and how they are defined (pixels, superpixels, channels, time–frequency bins).

5/ What is the complexity of ApprFIRM‑quantile and ApprFIRM‑kernel versus baselines?

---

### Official Review · Reviewer_pnkx · 2025-10-25

**Soundness:** 2
**Presentation:** 3
**Contribution:** 2
**Rating:** 2
**Confidence:** 4

**Summary:**

The paper proposes a global model-agnostic feature importance method based on conditional explanations. The method is based on FRIM and two novel approximations to feature importance. Results on two synthetic and one real-world EEG dataset demonstrate that the method results in better feature importance results in the presence of suppressor variables.

**Strengths:**

The paper is clearly written and easy to understand. The subfield of model explanation and feature importance is is a relevant field and the problems of existing methods pointed out in (and tackled by) the paper are important.

**Weaknesses:**

1. The proposed method is a global feature importance method but the comparison is made with permutation-based feature importance and two methods that were developed primarily for local feature importance. Model-specific feature importance is left out (arguably, because the goal is to be model-agnostic), wrapper based methods are left out (I assume because this is assumed to be to computationally intensive), but why aren't methods like Conditional Permutation Importance, Accumulated Local Effects (both are cited),  and the work on conditional Shapley values (Aas, K., Jullum, M., & Løland, A. (2021). Explaining individual predictions when features are dependent: More accurate approximations to Shapley values; Rozenfeld, I. (2024). Causal analysis of Shapley values: Conditional vs. marginal, and references therein).

2. Algorithm 1 appears to be highly sensitive to the choice of K. If K is 1, all features will have 0 importance (assuming that we just say that with a single observation we have 0 variability). If K = n (number of instances), all features will have the same non-zero importance (unless all predictions are the same). It is not clear that a robust choice of K can be made. Intuitively, Algorithm 2 suffers from the same issue with its bandwidth parameter.

3. The potential issue of Algorithm 1 when we have discrete features is acknowledged as a limitation, but the way the algorithms are defined, they only work with numerical features. That is, it is not clear how quantiles or distance is defined if the feature is categorical.

4. The experiments appear weak (also see Questions).

- It is great that the authors include a real-world example, but I think the authors should be more careful when drawing conclusions from these results. There is no ground truth to what the feature importance should be here, so we rely on the argument that feature importances produced by the proposed approach are more in line with what has been observed in physiological studies. This relies on the dataset being representative of this physiological phenomena AND that the models correctly learned the dataset. For example, if I made the claim that all the other methods in fact correctly reflect what the models have learned, but the proposed approach incorrectly excludes the mid-back area, there is nothing in the paper that would without a doubt convince me that that is not the case.

- The issue from above also applies to the synthetic datasets. It would be nice to have some evidence that the models have indeed learned the underlying concepts correctly. Maybe the models actually use features x2 and x3 and the competing methods are correct.

Minor issues:
* Paragraph starting at line 284: coef_x could be formatted better, especially in the non-bolded case it looks like coe f_s (coe of function f_s).
* The Algorithms were, at least for me, detailed enough so I could understand the method, so the text was more or less redundant. Also, the equations in text (lines 202, 214, etc.) result in extra spacing, which doesn't look very nice.
* Upper case typos in references (Erp, xai).

**Questions:**

* What is the purpose of Section 3.3? Isn't ApprFIRM already defined in Algorithm 1 and Algorithm 2?
* Assuming that the proposed method is, due to its design, superior in cases where there are suppressing variables, is there and downside/tradeoff that comes with that? That is, does the method then fail in some other cases or is it strictly superior?
* Experiments, 4. 1. 1: "are assigned a value of 1 for the positive class and -1 for the negative class, while all other features are initialized to 0". This reads like features x1 and x4 have only 2 possible values (which comes with other issues, such as x2 and x3 then not being irrelevant). Or does it refer to the means? This part would really benefit from a more clear description.
* Experiments: With 5 features, many otherwise computationally infeasible approaches could be applied to accurately determine feature importance, so I assume the reasoning here is that the method would help us with datasets that have lots of features but also feature relationships like these. But how realistic are such datasets that have such a specific relationship, lots of features, and that not only a handful of features are relevant anyway?

---

### Official Review · Reviewer_Lnbv · 2025-10-27

**Soundness:** 1
**Presentation:** 3
**Contribution:** 1
**Rating:** 2
**Confidence:** 2

**Summary:**

The paper proposes a model-agnostic, global feature-importance measure. Two practical approximations are presented: (i) a quantile-based estimator that averages model outputs within bins of $X_{S}$, and (ii) an RBF kernel–based estimator that reweights observed samples by proximity along the $X_{S}$​ axis. The authors argue these estimators reduce artifacts from marginal resampling and better handle correlated features and suppressor variables.

**Strengths:**

1. Recasting global importance as the dispersion of conditional means is a clean, intuitive framing.

2. The paper keeps the estimator model-agnostic and simple to implement

3. High-level motivation is easy to follow, and the exposition of the two estimators is accessible.

**Weaknesses:**

(Administrative) The submission includes a **non-anonymized GitHub link**, which violates the double-blind policy. This alone warrants reject.

Below are the technical weaknesses.

1. Overstated conditional-assumption. The quantile estimator implicitly assumes that samples within the same $X_{s}$ bin share (approximately) the same conditional distribution $P(X_{-s}| X_s}$. This is a strong—and often false—assumption in multimodal or highly interactive settings; two points with the same quantile of $X_{s}$ can correspond to very different joint contexts in $X_{-s}$. The paper provides neither a theoretical justification (e.g., conditions under which the bin-averaged estimator is consistent) nor empirical diagnostics (e.g., per-quantile covariance/mixture counts) to verify when the assumption holds.

2. Lack of a unifying view between the two estimators. The kernel approximation and the quantile approximation are presented as unrelated heuristics. There is no analysis of when one converges to the other, or shared conditions for consistency. Without this, the method family feels ad hoc rather than principled.

3. Correlation is not actually modeled—only avoided. While the approach avoids off-manifold combinations, it does not explicitly model  $P(X_{-s}| X_s}$. Weighting by distance *only in* $X_{s}$ ignores structure in $X_{-s}$ (e.g., multimodality conditional on $X_{s}$), so the claim that correlations are “handled” is, at best, relative to marginal resampling.

4. “Model-agnostic” claim not substantiated on images. If the method is truly model-agnostic, a low-dimensional image test (e.g., Fashion-MNIST) with pixel- or patch-level features should be feasible. No such evaluation is provided; this undermines generality claims beyond small tabular/EEG settings.

5. Figure readability and metrics. Figure 3 is hard to interpret: ground truth (or at least known signal-bearing features/regions) is not overlaid. For real data, quantitative metrics are absent—only qualitative maps are shown—so it is impossible to judge faithfulness, stability, or runtime trade-offs.

**Questions:**

See weaknesses above.

---

### Note · Authors · 2025-11-20

**Comment:**

We sincerely thank the reviewers for their thoughtful and valuable feedback. After careful consideration, we have decided to withdraw our current submission in order to focus on thoroughly revising and polishing our work, incorporating the reviewers' suggestions. We greatly appreciate the time and effort the reviewers have dedicated to our manuscript.

**Withdrawal Confirmation:**

I have read and agree with the venue's withdrawal policy on behalf of myself and my co-authors.